# Stem Cell-Derived Viral Antigen-Specific T Cells Suppress HIV Replication and PD-1 Expression on CD4+ T Cells

**DOI:** 10.3390/v13050753

**Published:** 2021-04-25

**Authors:** Mohammad Haque, Fengyang Lei, Xiaofang Xiong, Yijie Ren, Hao-Yun Peng, Liqing Wang, Anil Kumar, Jugal Kishore Das, Jianxun Song

**Affiliations:** 1Department of Microbial Pathogenesis and Immunology, Texas A&M University Health Science Center, Bryan, TX 77807, USA; mhaque@pennstatehealth.psu.edu (M.H.); xxiong@tamu.edu (X.X.); ren@tamu.edu (Y.R.); haoyun0122@tamu.edu (H.-Y.P.); wang1989@tamu.edu (L.W.); anil85krp@tamu.edu (A.K.); jugal@tamu.edu (J.K.D.); 2Department of Ophthalmology, Harvard University School of Medicine, Boston, MA 02215, USA; Dylan_Lei@MEEI.HARVARD.EDU

**Keywords:** HIV, stem cells, cell differentiation, T cells, viral antigen, viral replication, adoptive cell transfer, T cell exhaustion, mice

## Abstract

The viral antigen (Ag)-specific CD8+ cytotoxic T lymphocytes (CTLs) derived from pluripotent stem cells (PSCs), i.e., PSC-CTLs, have the ability to suppress the human immunodeficiency virus (HIV) infection. After adoptive transfer, PSC-CTLs can infiltrate into the local tissues to suppress HIV replication. Nevertheless, the mechanisms by which the viral Ag-specific PSC-CTLs elicit the antiviral response remain to be fully elucidated. In this study, we generated the functional HIV-1 Gag epitope SL9-specific CTLs from the induced PSC (iPSCs), i.e., iPSC-CTLs, and investigated the suppression of SL9-specific iPSC-CTLs on viral replication and the protection of CD4+ T cells. A chimeric HIV-1, i.e., EcoHIV, was used to produce HIV replication in mice. We show that adoptive transfer of SL9-specific iPSC-CTLs greatly suppressed EcoHIV replication in the peritoneal macrophages and spleen in the animal model. Furthermore, we demonstrate that the adoptive transfer significantly reduced expression of PD-1 on CD4+ T cells in the spleen and generated persistent anti-HIV memory T cells. These results indicate that stem cell-derived viral Ag-specific CTLs can robustly accumulate in the local tissues to suppress HIV replication and prevent CD4+ T cell exhaustion through reduction of PD-1 expression.

## 1. Introduction

While antiretroviral therapy (ART) in human immunodeficiency virus (HIV) infection is available and new antiviral drugs are being developed, elimination of persistently infected cells is still a major issue. Highly active ART (HAART) consists of the combination of > three antiretroviral (ARV) medications to greatly suppress the HIV virus and prevent the disease progression from acute HIV infection, and chronic HIV infection to acquired immune deficiency syndrome (AIDS). Enormous reductions have been observed in rates of death and suffering when use is made of a potent ARV regimen, particularly in the early stages of the disease. However, there is still no cure for HIV infection, given the promises utilizing HAART and/or stem cell transplantation [1,2]. T cells play a crucial role in HIV infection; a robust T-cell response is associated with control of HIV infection; however, HIV-specific T cells are deleted or dysfunctional in HIV/AIDS [3,4,5]. As a result, current HAART treatment can control/eliminate the virus but cannot restore T cell immunity.

Adoptive cell transfer (ACT) of CD8^+^ cytotoxic T lymphocytes (CTLs) with specificity for HIV antigen (Ag) ^+^ cells represents an approach aiming to ultimately eliminate infected cells such as CD4^+^ T cells, macrophages (M_Φ_), and dendritic cells (DC) carrying HIV gene [6]. HIV can remain a chronic, persisting infection due to its ability to stay hidden within infected blood cells. These cellular “HIV reservoirs” contain the genetic code of HIV. The viruses remain invisible to the body’s immune defenses and are not sensitive to anti-HIV drugs by escape mutations [7]. Optimal CTLs home to sites of infection, directly lyse infected cells, secrete cytokines that recruit other immune cells, and develop into long-lived memory cells that control HIV immune escape [8,9]. ACT of CTLs recognizing a few viruses in immune-compromised patients (e.g., CMV) has great promise in reconstituting anti-viral immunity [10,11]. Efforts to use the same strategy against HIV infection in patients with AIDS have so far been met with limited success. However, as in other chronic persistent infections such as hepatitis B virus (HBV), T cells appear to play a crucial role in HIV infection. Ultimately, definitive cures for HIV/AIDS need both control/elimination of the virus (HIV reservoirs) and restoring T cell immunity. A combination of ART and ACT of T cells may result in a cure of HIV/AIDS. For ACT-based immunotherapy, the in vitro generation of viral Ag-specific T cells for in vivo re-infusion is an optimal approach [12,13,14,15]. However, current methodologies are limited in terms of the capacity to generate, isolate, and expand enough of such T cells from patients for therapeutic interventions. 

In this study, we develop a practical system to generate HIV-1 Gag epitope SL9-specific CTLs from induced pluripotent stem cells (iPSCs), i.e., iPSC-CTLs, which present the typical T cell features, including expressions of CD3, CD8 and T cell receptor (TCR), and production of cytokines including IFN-γ. We show that a chimeric HIV-1, EcoHIV, induced HIV replication in mice; adoptive transfer of SL9-specific iPSC-CTLs greatly suppressed HIV replication in the peritoneal macrophages and spleen in the animal model. We demonstrate that the adoptive transfer significantly reduced expression of PD-1 on CD4^+^ T cells in the spleen. These results indicate a useful regimen of HIV infection through ACT of stem cell-derived viral Ag-specific CTLs.

## 2. Results 

### 2.1. Generation of HIV-Specific iPSC-CTLs

We used the MiDR retrovirus-mediated transduction and transduced mouse iPSCs with a human-mouse hybrid SL9-specific TCR (MiDR-SL9-TCR, Figure 1A) [16,17] or OVA_257–264_ TCR (MiDR-OVA TCR), followed by a co-culture with OP9 stromal cells expressing Notch ligands DL1 and DL4 (OP9-DL1/DL4) in the presence of recombinant cytokines of rIL-7 and rFlt3L. Upon gene transduction, we visualized the DsRed expression by fluorescence microscopy (Figure 1B), and sorted GFP^+^DsRed^+^ cells. After 8 days of culture with the OP9-DL1/DL4 cells, iPSCs differentiated into mesoderm-like cells, and showed characteristic, non-adherent grape-like cluster morphology on day 16. On day 24, lymphocyte-like cells spread fully on the culture plates (Figure 1C). On day 28 of in vitro co-culture, the iPSC-derived cells substantially expressed CD3 and CD8, the T cell markers. Flow cytometric analysis of CD3^+^CD8^+^ populations showed that SL9 but not OVA TCR transduction substantially resulted in the generation of SL9-specific CD8^+^ T cells (CD8^+^ SL9 TCR^+^; Figure 1D). These results suggest that iPSCs can differentiate into HIV Ag-specific CD8^+^ T cells by the approach of TCR gene transduction, followed by an in vitro stimulation with stromal cells expressing Notch ligands.

To determine the functional status of SL9-specific iPSC-CTLs, we tested whether these iPSC-CTLs had the capacity to produce antiviral cytokines, following viral Ag stimulation. On day 28 of in vitro co-culture, we isolated the CD4^−^CD8^+^ single-positive (SP) iPSC-CTLs and stimulated by T-depleted splenocytes pulsed with SL9 peptide and assessed cytokine production. The iPSC-CTLs produced large amounts of IL-2 and IFN–γ as detected by intracellular staining (Figure 2A) or ELISA (Figure 2B), and displayed Ag-specific cytotoxicity (Figure 2C), which were similar as SL9 TCR gene-transduced CTLs (All *p* > 0.05; multiple t tests between SL9-specific iPSC-CTLs and SL9-specific CTLs). These results confirmed the generation of functional HIV viral Ag-specific iPSC-CTLs by a combination of TCR gene transduction and in vitro stimulation of Notch signaling.

### 2.2. EcoHIV Replication in Mice 

To demonstrate the protection of HIV infection by adoptive cell transfer of HIV viral Ag-specific iPSC-CTLs, we induced EcoHIV infection in HLA-A2.1 transgenic (HHD) mice by the intraperitoneal injection (*i.p.*) with EcoHIV virus. The EcoHIV chimeric virus contains all the known coding and regulatory regions of the HIV-1 genome except for gp120, and gp41 is unlikely to be expressed because it lacks an in-frame codon for initiation of translation. The biological activity of this virus has been tested by several approaches [18]. Three weeks after the EcoHIV infection, mice were euthanized, and tissues were collected for analysis of viral replication by qPCR. We did not detect any HIV DNA copy in uninfected mice, but substantially revealed in the spleen, liver, and kidney of EcoHIV-infected mice (Figure 3A). In addition, after we confirmed viral infection in mice, we investigated the impact of EcoHIV on T cells. One million cells were isolated from the spleen from each sample and cells were analyzed for GFP^+^ CD4 and CD8 by flow cytometry. We did not observe any difference in the numbers of CD4^+^ and CD8^+^ T cells after 2 weeks of infection but did detect a dramatic decrease in number of CD4^+^ T cells after 8 weeks of infection (Figure 3B). These results indicate that EcoHIV infection in HHD mice caused viral DNA replication in various tissues and reduced the overall number of CD4^+^ T cells during chronic infection.

### 2.3. Adoptive Transfer of SL9-Specific iPSC-CTLs Reduced EcoHIV Replication

EcoHIV replication can be tracked underlying its GFP expression [19]. To determine the effect of adoptive transfer of SL9-specific iPSC-CTLs on HIV replication, we used the IVIS Lumina series III imaging system to visualize EcoHIV replication in vivo. The series III imaging system provides the highest degree of sensitivity of any optical imaging system. Mice were adoptively transferred with SL9-specific iPSC-CTLs or control cells and were administrated for EcoHIV infection 1 weeks later. Three weeks after EcoHIV infection, we anaesthetized mice and placed under the Lumina system. We could clearly distinguish the GFP fluorescence at the site of EocHIV relocation. We detected the viral DNA replication in the spleen, liver, and kidney, and EcoHIV strongly replicated in the spleen (Figure 3A), which was also monitored by the IVIS Lumina system (Figure 4A). To further confirm this observation, we sacrificed all groups of mice and their spleens were isolated and placed under the Lumina to detect the GFP fluorescence. Our data obviously presented that the fluorescence occurring from the spleen as we detected several fluorescent areas in the spleens from the mice infected with EcoHIV but not uninfected animals (Figure 4B). In these studies, adoptive transfer of SL9-specific iPSC-CTLs reduced EcoHIV replication as evidenced by the reduction of GFP fluorescence from the spleens in mice receiving EcoHIV infection and the adoptive transfer compared to mice with EcoHIV infection but with the control cell transfer (Figure 4A,B). As peritoneal macrophages are the most attractive target for HIV infection. We also examined the EcoHIV replication in the peritoneal macrophages. After the adoptive transfer and EcoHIV infection, we sacrificed the mice and collect the peritoneal macrophages from all groups of mice. Macrophages were prepared for confocal microscopy to detect the expression of GFP. Our observation undoubtedly showed that EcoHIV replication was also reduced in mice receiving SL9-specific iPSC-CTLs compare to those mice with the control cell transfer (Figure 4C). These imaging results suggest that viral DNA replication was reduced after the adoptive transfer of SL9-specific iPSC-CTLs.

To directly examine viral DNA replication, we performed qPCR. Three weeks after EcoHIV infection, mice were sacrificed and their spleen, liver, kidney tissues and peritoneal macrophages were isolated. DNA was isolated from all infected samples and macrophages. DNA was used to detect the virus replication. Our qPCR data evidently showed that virus replications were reduced in mice receiving SL9-specific, but not non-specific iPSC-CTLs (Figure 5A). We also monitored GFP expression in the spleen, representing the presence of EcoHIV and an alternative signal of EcoHIV replication. Single cell suspensions were made from the spleens and cells were analyzed for expression of GFP by flow cytometry. Our results visibly showed that GFP expression was observed in the spleens of mice infected with EcoHIV and reduced in those of mice receiving SL9-specific iPSC-CTLs (4.82% vs. 1.02%; Figure 5B). In addition, we performed immunohistochemistry staining of spleen samples to reveal EcoHIV-infected GFP^+^ cells. We were able to detect a significant number of GFP^+^ cells in EcoHIV-infected mice receiving control cells, and the GFP^+^ cells were substantially reduced in mice receiving SL9-specific iPSC-CTLs (Figure 5C,D). Collectively, these results suggest that adoptive transfer of SL9-specific iPSC-CTLs can suppress viral DNA replication.

### 2.4. SL9-Specific iPSC-CTLs Suppressed PD-1 Expression on CD4^+^ T Cells 

PD-1 is T-cell exhaustion marker, and its expression on CD4^+^ T cells is increased during chronic HIV infection [20]. To determine the mechanisms by which adoptive transfer of SL9-specific iPSC-CTLs can suppress viral DNA replication and to protect CD4^+^ T cells, the main virus-targeting cells, we analyzed PD-1 expression on CD4^+^ T cells from treated or untreated mice. Eight weeks after EcoHIV infection, single cell suspensions were made from the spleens and were analyzed expression of PD-1 on CD4^+^ T cells by flow cytometry. We found that PD-1 expression on CD4^+^ T cells was dramatically reduced from mice receiving SL9-specific iPSC-CTLs compared to those mice with the control cell transfer (8.73% vs. 2.23%; Figure 6A,B). This result strongly indicates that the suppression of HIV DNA replication by adoptive transfer of SL9-specific iPSC-CTLs protected cell exhaustion of CD4^+^ T cells through reduction of PD-1 expression.

### 2.5. SL9-Specific iPSC-CTLs Persist In Vivo

We further determined whether the adoptive transfer of SL9-specific iPSC-CTLs could generate T cell persistence that is critical for protecting against chronic HIV infection. We performed the adoptive transfer using SL9 or OVA_257-264_ TCR gene-transduced CD8^+^ T cells from the lymph nodes (LNs) and spleen of HHD mice or the Ag (SL9 or OVA)-specific mouse iPSC-CTLs into HHD mice. Thirty-five days later, the T cell persistence was analyzed by tracking CD8^+^CD44^+^ SL9 TCR^+^ cells from the pooled LNs and spleen. A greater number of CD44^+^ SL9 TCR^+^ persistent CD8^+^ T cells developed in mice receiving SL9-specific iPSC-CTLs than in the animals receiving TCR-transduced T cells (8.26% vs. 3.05%) in the pooled LNs and spleen, as analyzed by flow cytometric analysis (Figure 7A) and cell number (Figure 7B; *p* < 0.01). The control transfers with equal number of OVA-specific iPSC-CTLs or OVA TCR-transduced CTLs did not generate any obvious SL9-specific persistent T cells in the LNs and spleen (Figure 7A,B). These results suggest that SL9-specific iPSC-CTLs can generate T cell persistence. 

## 3. Discussion

In this study, we generated functional viral Ag-specific iPSC-CTLs and investigated their ability in suppressing HIV infection in a murine model. We show that adoptive transfer of SL9-specific iPSC-CTLs greatly suppressed EcoHIV replication in the host tissues or cells. Furthermore, we demonstrate that the adoptive transfer dramatically reduced expression of PD-1 on CD4^+^ T cells. These results indicate that stem cell-derived viral Ag-specific CTLs can suppress HIV replication and prevent T cell exhaustion by reducing PD-1 expression on CD4^+^ T cells. 

Virus-specific T cells capable of controlling HIV and AIDS are deleted or dysfunctional in patients with HIV infection. Current antiviral therapy (e.g., nucleoside reverse transcriptase inhibitors/NRITS, non-NRITS, protease inhibitors, fusion inhibitors, or other inhibitors) rarely establishes immunological control over HIV replication. ACT of HIV-specific CD8^+^ CTLs has a great potential to eradicate the HIV reservoirs; a combination of anti-HIV drugs and ACT of HIV-specific CTLs is likely to result in a cure of chronic HIV infection. Naive or central memory T cell-derived effector CTLs known as “highly reactive” CTLs, are optimal populations because these cells have a high proliferative potential, are less prone to apoptosis than terminally differentiated cells and have a higher ability to respond to homeostatic cytokines. However, such ACT has often not been feasible due to difficulties in obtaining enough cells from patients. 

Strong arguments support the development of adoptive T cell therapies using engineered T cells with specificity for virus-infected cells for the treatments of chronic HIV infection [21]. While clinical trials show safety, feasibility, and potential therapeutic activity of cell-based therapies using this approach, there are concerns that undesirable effects arising from autoimmunity due to cross-reactivity from mispairing TCR [22,23] or off-target Ag recognition by non-specific TCR [24,25] or on-target off toxicity by CAR [25,26] with healthy tissues could occur. In addition, genetically modified T cells using current approaches are usually intermediate or later effector T cells, which only have short-term persistence in vivo. To date, PSCs are the only source available to generate a high number of naive single-type Ag-specific T cells [16,27,28,29]. IPSCs can be easily generated from patients’ somatic cells by transduction of various transcription factors, and exhibit characteristics identical to those of ESCs. Due to the plasticity and the potential for an unlimited capacity for self-renewal, iPSC therapies have great potential in regenerative medicine and tissue replacement after injury or disease. In addition, iPSCs have high potential for advancing the field of cell-based therapies [30]. Our laboratory has published results indicating that reprogramming of Ag-specific CTLs or regulatory T cells (T_regs_) from iPSCs (i.e., iPSC-CTLs, iPSC-T_regs_) can be used for cell-based therapies [16,27,31,32]. Using our programming approach that is not complex and expensive as compared to others [28,29], we have developed a system to generate HIV-specific iPSC-CTLs. Our current studies validate this system and provide new insight into the methodologies and mechanistic requirements for efficient development of viral Ag-specific PSC-CTLs. Once such strategies become available, there is the potential to facilitate the generation of immune surveillance. Thus, important advances in cell-based therapies of chronic HIV infection are anticipated from the proposed studies.

Taken together, the current study provides new insights into the therapeutic intervention using viral Ag-specific iPSC-CTLs as immunotherapy for HIV infection. However, in the current study, the EcoHIV infection in mice is not a real model of chronic HIV infection. The robust animal models for studying HIV persistence and therapeutic intervention may be humanized NSG mice. NSG mice are readily engraft both human peripheral blood mononuclear cells (PBMCs) and human hematopoietic stem cells (HSCs) to create humanized NSG mice. The engrafted human cells differentiate and mature into the full spectrum of human immune cells and are capable of a wide range of human immune functions. As such the humanized NSG mice have developed into a much-needed model for HIV/AIDS and other infectious disease research [33]. Our results need to be validated by the humanized NSG mice that are infected with HIV virtues. In addition, the mechanisms by which the viral Ag-specific PSC-CTLs elicit the antiviral response remain to be fully elucidated.

## 4. Material and Methods

### 4.1. Ethics Statement

All experiments were approved and performed in compliance with the regulations of The Texas A&M University Animal Care Committee (IACUC; #2018-0006; 03/13/2018) and in accordance with the guidelines of the Association for the Assessment and Accreditation of Laboratory Animal Care.

### 4.2. Cell Lines and Mouse 

Mouse iPS-MEF-Ng-20D-17 cell line was obtained from the RIKEN Cell Bank [16]. The OP9-DL1/DL4 cell line was generated by a retroviral transduction of the OP9 cells [34]. SNL76/7 cell line (ATCC^®^ SCRC-1049^™^) was purchased from ATCC (Manassas, VA, USA). H-2 class I knockout, HLA-A2.1-transgenic (HHD) mice were obtained from Dr. Francois A. Lemonnier (The Pasteur Institute, Paris, France) [31]. 

### 4.3. Cell Culture 

iPSCs were maintained on feeder layers of irradiated SNL76/7 cells in 6-well culture plates (Nunc), and were passaged every 3 days [16]. In brief, iPSCs were maintained in DMEM culture medium supplemented with 15% fetal calf serum (FCS), 0.1 mmol/L nonessential amino acids, 1 mmol/L L-glutamine (All were from Invitrogen), and 0.1 mmol/L β-mercaptoethanol (Sigma-Aldrich, St. Louis, MO, USA). Monolayers of OP9-DL1/DL4 cells were cultured in α-MEM medium supplemented with 20% FCS and 2.2 g/L sodium bicarbonate (All were from Invitrogen). The iPSCs were washed once in the OP9-DL1/DL4 medium before plating onto sub-confluent OP9-DL1/DL4 monolayers for T lineage differentiation in the presence of murine recombinant Flt-3 ligand (mrFlt3L; 5 ng/mL; Peprotech, Rocky Hill, NJ, USA) and 1 ng/mL murine IL-7 (mIL-7; Peprotech).

### 4.4. EcoHIV Infection

EcoHIV is a chimeric HIV-1 expressing enhanced green fluorescent protein (EGFP) as an indicator, which was constructed on the backbone of HIV-1/NL4-3, as described previously (1). Infectious EcoHIV stocks were propagated in HEK293TN cells as described (4) and tittered for the p24 HIV-1 core Ag by ELISA, following the manufacturer’s instructions (ZeptoMetrix Corporation, NY, USA). Each mouse was i.p. injected with 3 × 10^6^ pg of EcoHIV virus.

### 4.5. Retroviral Transduction and Generation of Ag-Specific iPSC-CTLs 

cDNA for SL9 (HIV-1 Gag_77-85_; SLYNTVATL)-specific human-murine hybrid TCR genes or OVA_257–264_ (SIINFEKL)-specific H-2K^b^-restricted TCR genes (Vα2 and Vβ5) kindly provided by Dr. Harris Goldstein (Albert Einstein College of Medicine) and Dr. Dario A. Vignali (University of Pittsburgh, PA, USA) was sub-cloned in the MIDR construct (DsRed^+^) and used for retroviral transduction of mouse iPSCs (GFP^+^) and generation of SL9 or OVA-specific iPSC-CTLs [34]. CD8^+^ T cells transduced with the SL9-specific TCR can be specifically activated and have a strong killing effect on target cells in HIV immunotherapy [17,35,36]. Retroviral transduction was performed as described previously [37]. The iPSCs were grown on feeder layers of irradiated SNL76/7 cells. After 2 days, the supernatant was replaced with 1 mL viral supernatant containing 5 µg/mL Polybrene (Sigma-Aldrich), and the cells were spun for 1 h at 32 °C and incubated at 32 °C for 8 h. This procedure was repeated the following day. Viral supernatant was removed and replaced with fresh medium, and the iPSCs were re-cultured. Expression of DsRed was determined by flow cytometry gating on GFP^+^ cells. DsRed^+^ GFP^+^ cells were purified from cell sorting using a MoFlo high performance cell sorter (Beckman Coulter, Fullerton, CA, USA).

### 4.6. Antibodies

PE-, PE/Cy7, Alexa 647, APC or APC/Cy7-conjugated anti-mouse CD3 (17A2), CD8 (53-6.7), CD4 (GK1.5), IFN–γ(XMG1.2), CD44 (IM7), PD-1 (29F.1A12) and anti-human TCRα/β (IP26) were obtained from BioLegend (San Diego, CA, USA). HLA-A201-HIV-1_77-85_-PE tetramer (MHC-LC396) was purchased from Creative BioLabs (Shirley, NY, USA).

### 4.7. Flow Cytometric Analysis

Gene-transduced iPSCs were cocultured with OP9-DL1/DL4 cells for various days, and the expression of CD3, TCR, and CD8 was analyzed by flow cytometry after gating on DsRed^+^ cells or other markers such as CD3 or TCR. T cells from the spleen was collected and analyzed by flow cytometry for CD3, CD4, CD8 or PD-1.

### 4.8. Adoptive Cell Transfer

Six-week-old HHD mice were i.v. injected with DsRed^+^CD8^+^ T progenitors (2 × 10^6^) from iPSCs (in PBS), and one week later were i.p. administrated with EcoHIV viruses. At the indicate time points, mice were sacrificed, and the spleens, livers and kidneys were removed for histopathological examination.

### 4.9. QPCR

QPCR quantitation of total HIV DNA in EcoHI-infected mice and DNA standardization was described [38]. In brief, EcoHIV DNA from the *gag* gene region was detected with forward primer 5′–TGGGACCACAGGCTACACTAGA–3′, reverse primer 5′–CAGCCAAAACTCTTGCTTTATGG–3′ and probe 5′–TGATGACAGCATGCCAGGGAGTGG–3′.

### 4.10. Confocal Microscopy

Microscopy was conducted as described [19]. Briefly, Mice were perfused first with phosphate-buffered saline (PBS) followed by 4% paraformaldehyde, tissues were then dissected, flash-frozen, and stored at –80 °C. Thirty-micrometer-thick coronal sections were cut on a Leica cryostat and stored in cryoprotectant at −20 °C until staining. For immunofluorescence staining, sections were rinsed in PBS, treated with 10% fetal bovine serum (FBS) and 0.2% Triton X-100 in PBS for 1 h, washed 3 times in PBS-0.2% Triton X-100, and incubated with chicken anti-GFP (1:500; Life Technologies) (for the detection of the marker protein in EcoHIV) at 4 °C overnight. Sections were then washed and incubated with goat anti-chicken Alexa 488 (1:100; Life Technologies) in PBS-10% FBS + 0.2% Triton X-100, at 37 °C for 2 h, washed, and mounted onto slides with PROLONG GOLD ANTIFADE Reagent. Images including Z-stacking were captured with fully motorized Leica TCS FP5 confocal microscope and analyzed using Improvision Velocity software (Version X; PerkinElmer).

### 4.11. Live Mice Imaging

At different days of post infection of EcoHIV, mice were anesthetized and GFP expression was visualized by IVIS Lumina series III imaging system (PerkinElmer). The series III imaging system provides the highest degree of sensitivity of any optical imaging system. To confirm the location of EcoHIV in mice, mice were sacrificed, and spleen tissues were isolated. Isolated spleen tissues were again visualized through IVIS Lumina series III imaging system.

### 4.12. Statistical Analysis

The unpaired t test was performed for analysis of the differences between the groups, using the GraphPad Prism 9 (GraphPad Software, San Diego, CA, USA), and the significance was set at 5%.

## Figures and Tables

**Figure 1 viruses-13-00753-f001:**
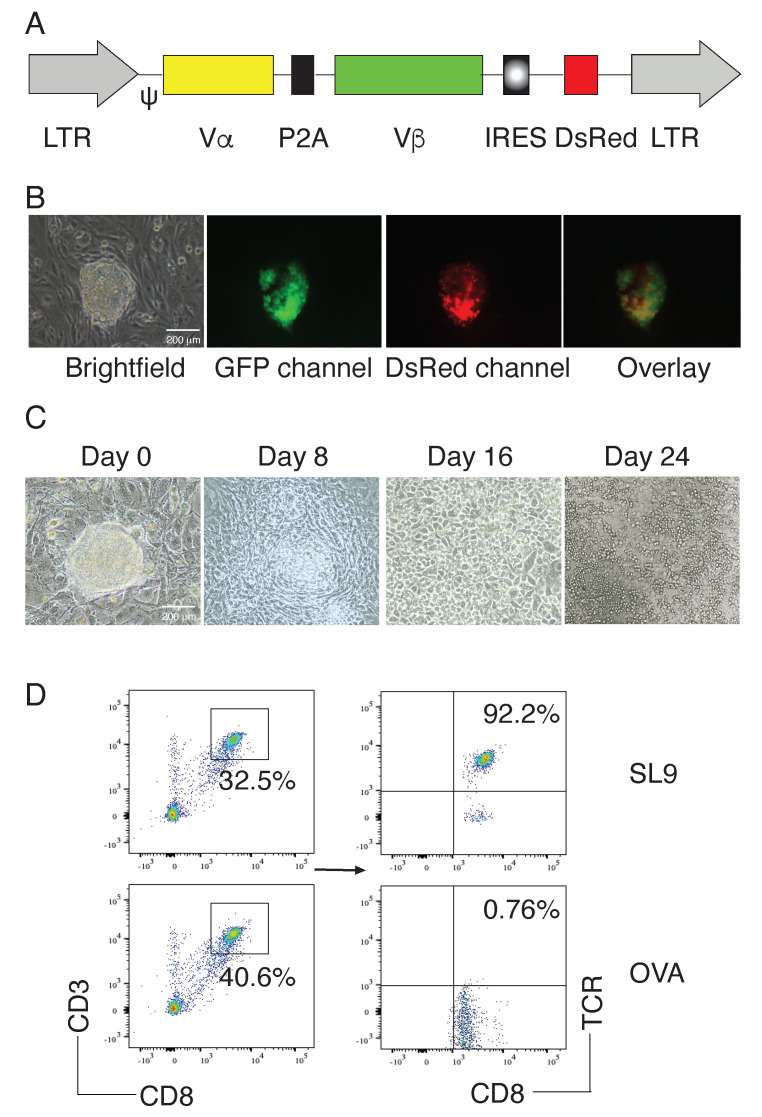
Generation of SL9-specific iPSC-CTLs. Mouse iPSCs were transduced with the following retroviral constructs: SL9 TCR (MiDR-SL9 TCR) or OVA_257–264_ TCR (MiDR-OVA TCR), and the transduced iPSCs were co-cultured with OP9-DL1/DL4 stromal cells for T lineage differentiation. (**A**) Schematic representation of the retrovirus constructs expressing HIV SL9 TCR. Ψ, packaging signal; 2A, picornavirus self-cleaving 2A sequence; LTR, Long terminal repeats. (**B**) The SL9 TCR-transduced iPSCs were visualized by a fluorescence microscope. (**C**) Morphology of T cell differentiation on various days. (**D)** Flow cytometric analysis of the iPSC-derived cells on day 28. CD3^+^CD8^+^ cells were gated as indicated and analyzed for the expression of CD8 and SL9 TCR. Data shown are representative of three identical experiments.

**Figure 2 viruses-13-00753-f002:**
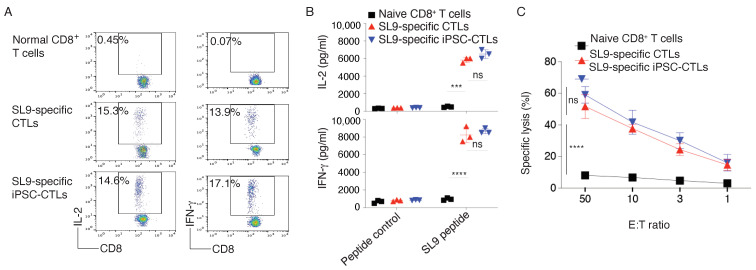
Functional analysis of SL9-specific iPSC-CTLs. On day 28 of in vitro co-culture, the SP CD8^+^ iPSC-T cells were sorted. The iPSC-T cells and CD8^+^ T cells transduced with MiDR-SL9 TCR were stimulated by T-depleted splenocytes (APCs) from HHD mice and pulsed with SL9 peptide (SLYNTVATL). (**A**) Intracellular staining of IL-2 and IFN-γ after 7 h (gated on CD8^+^ cells) (T/APCs = 1:4). Data shown are representative of three individual experiments (*n* = 3). (**B**) ELISA of IL-2 and IFN-γ after 40 h. The values represent mean ± S.E.M. (***, *p* < 0.001, ****, *p* < 0.0001; ns, *p* > 0.05. Unpaired t tests). (**C**) T cell cytotoxicity was measured after co-culture for 6 h using the 7-AAD/CFSE cell-mediated cytotoxicity assay kit. The values represent mean ± S.E.M. (****, *p* < 0.0001; ns, *p* > 0.05. Nested one-way ANOVA).

**Figure 3 viruses-13-00753-f003:**
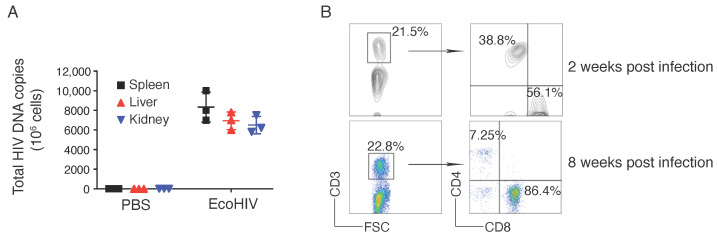
EcoHIV infection in mice. HHD mice were *i.p.* administrated with EcoHIV viruses. (**A**) Total HIV DNA copies. Three weeks after EcoHIV infection, the tissues of spleen, liver, kidney were isolated, and DNA was extracted for qPCR. Data shown are three individual experiments (*n* = 5). The line represents mean values. (**B**). Analysis of CD4^+^ T cells in the spleens after EcoHIV infection. In weeks 2 and 8 after EcoHIV infection, single cell suspensions were made from the spleens and cells were analyzed for CD3 (left), and CD4 with CD8 were analyzed (right), gating on CD3^+^ populations by flow cytometry. Data are representative of three independent experiments (*n* = 5).

**Figure 4 viruses-13-00753-f004:**
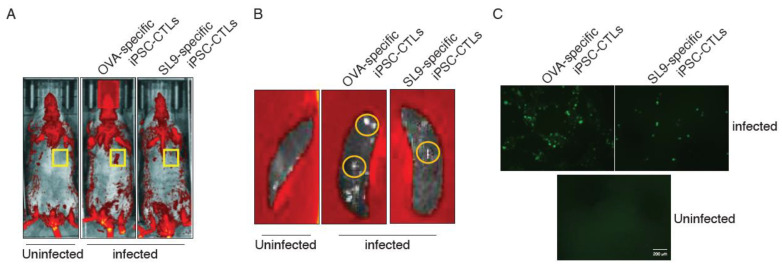
Adoptive transfer of SL9-specific iPSC-CTLs suppress EcoHIV infection in the spleens. HHD mice were i.v. adoptively transferred with SL9 or OVA_257–264_-specific DsRed^+^CD8^+^ pre-iPSC-CTLs and in the following week were i.p. infected with EcoHIV. Three weeks after EcoHIV infection, EcoHIV infection was analyzed. (**A**) Mice were monitored under the Lumina system. EcoHIV GFP fluorescence (yellow squares) was visualized. (**B**) Spleens were monitored under the Lumina system. EcoHIV GFP fluorescence (yellow cycles) was visualized. (**C**) Peritoneal macrophages were detected expression of GFP under a confocal microscope. All data are representative of five mice per group of three independent experiments.

**Figure 5 viruses-13-00753-f005:**
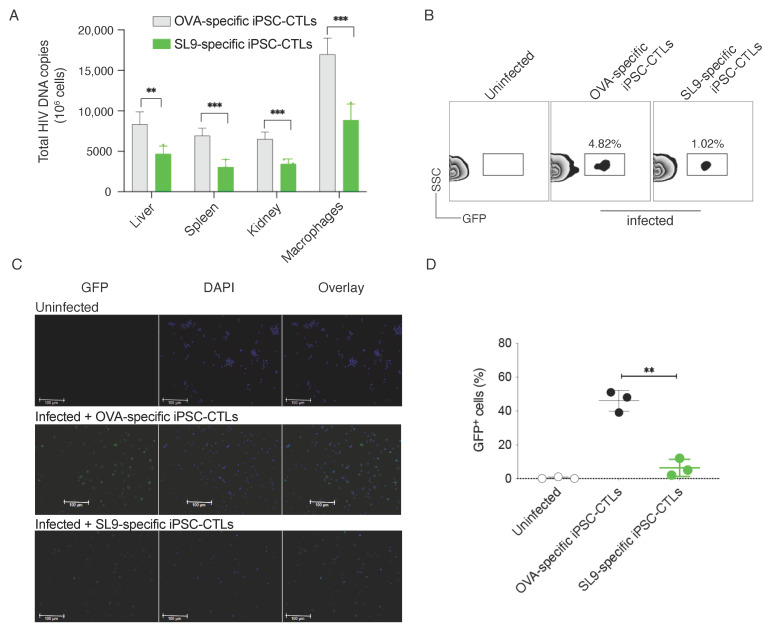
Adoptive transfer of SL9-specific iPSC-CTLs suppress EcoHIV replication. HHD mice were *i.v.* adoptively transferred with SL9 or OVA_257–264_-specific DsRed^+^CD8^+^ pre-iPSC-CTLs and in the following week were *i.p.* infected with EcoHIV. Three weeks after EcoHIV infection, EcoHIV replication was analyzed. (**A**) qPCR. DNA from mouse spleen, liver, kidney tissues and peritoneal macrophages was analyzed for viral replication by qPCR. (**B**) EcoHIV replication was monitored by GFP expression in the spleens by flow cytometry. (**C**) EcoHIV GFP^+^ cells were examined by immunohistochemistry staining of spleens. Data are representative of five mice per group of three independent experiments. (**D**) Quantification of GFP^+^ T cells. The values represent mean ± S.E.M. (** *p* < 0.01, *** *p* < 0.001, Unpaired t test).

**Figure 6 viruses-13-00753-f006:**
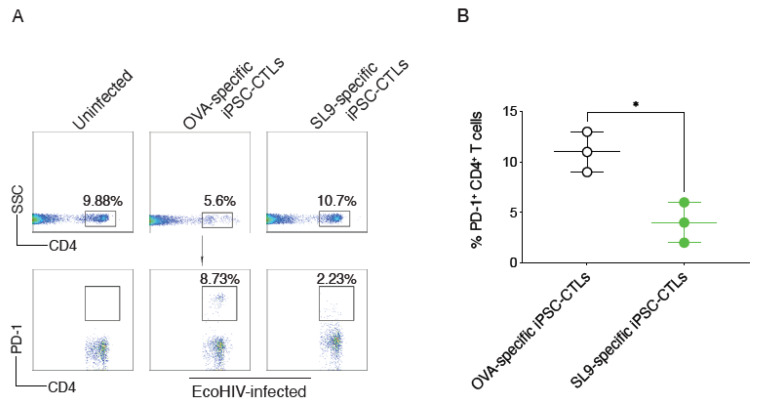
Adoptive transfer of SL9-specific iPSC-CTLs suppress PD-1 expression on CD4+ T cells. HHD mice were *i.v.* adoptively transferred with SL9 or OVA_257–264_-specific DsRed^+^CD8^+^ pre-iPSC-CTLs and in the following week were *i.p.* infected with EcoHIV. Eight weeks after EcoHIV infection, single cell suspensions were made from the spleens. (**A**) Expression of PD-1 on CD4^+^ T cells was determined by flow cytometry. Data shown are the representative of three identical experiments (*n* = 5). (**B**) Quantification of PD-1^+^ CD4^+^ T cells. The values represent mean ± S.E.M. (* *p* < 0.05, Unpaired t test).

**Figure 7 viruses-13-00753-f007:**
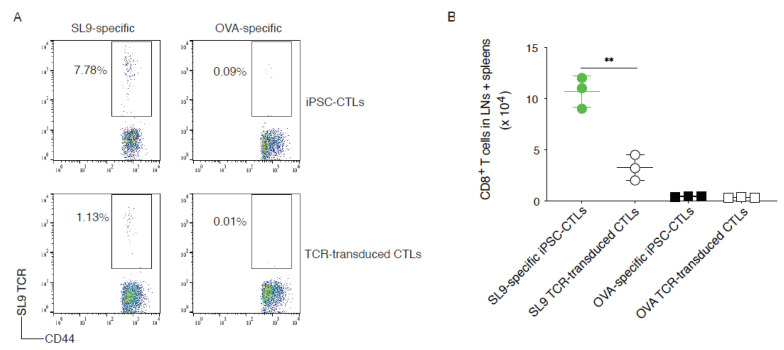
SL9-specific iPSC-CTLs persist in vivo. HHD mice were *i.v.* adoptively transferred with SL9 or OVA_257–264_-specific DsRed^+^CD8^+^ pre-iPSC-CTLs and in the following week were *i.p.* infected with EcoHIV. Eight weeks after EcoHIV infection, single cell suspensions were made from the spleens. (**A**) The spleen was analyzed for the T cell persistence. Persistent T cells by flow cytometry using CD44 and anti-human TCR staining, gating on CD8^+^ populations. Data shown are the representative of three identical experiments (*n* = 5). (**B**) Number quantification of persistent CD8^+^ T cell population. The values represent mean ± S.E.M. (** *p* < 0.01, Unpaired t test).

## Data Availability

The data presented in this study are available on request from the corresponding author.

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
