# Peer review of "Stem Cell-Derived Viral Antigen-Specific T Cells Suppress HIV Replication and PD-1 Expression on CD4+ T Cells"

_viruses, 2021, doi:10.3390/v13050753_

Round 1
Reviewer 1 Report
This paper develops and adapts previous work by the group in other viral systems to application of the adoptive transfer of antigen specific iPSC to control of HIV-1 infection in a murine model. The study represents a robust proof of principle supporting further development of the system towards therapeutic use.
There are several clarification of the model system and the data presentation, which should be addressed.
- The author introduce the manuscript in discussing the therapeutic potential/ advantage of an adoptive transfer system in HIV-1 infection (capturing virus emergent from reservoirs for example), whereas the model as described is entirely prophylactic. A. what is the impact of adoptive transfer of iPSC-CTL if performed after EcoHIV infection?
- The authors detect a high frequency of IL-2 production within their SL9-specifc CTL and iPSC-CTL. Is this ever observed in untransduced antigen specific normal CD8+ T cells in the EcoHIV model? In the model shown how are transferred SL9-specifc iPSC-CTL maintained? Do they require autocrine IL-2 or is there a role for CD4+ t cell derived IL-2 after infection? What is the half-life of the transferred cells?
- The authors focus on suppression of HIV-1 replication in peritoneal macrophages. What is the impact on infected CD4+ T cells?
- In figures 2, 3, 6 and 7 can the authors clarify what is meant by ‘ 3 duplicate experiments’ 3 independent experiments etc and describe precisely what id represented by the bars around the individual values (median with IQR, mean with SD etc?). The precise number of mice tested in each group or replicate experiment should be stated.
5. In figure 5C the authors should either quantify the data in a graph or depict the number of GFP positive events on the figure.
Author Response
Q1. The author introduce the manuscript in discussing the therapeutic potential/ advantage of an adoptive transfer system in HIV-1 infection (capturing virus emergent from reservoirs for example), whereas the model as described is entirely prophylactic. A. what is the impact of adoptive transfer of iPSC-CTL if performed after EcoHIV infection?
Answer: We are working on the therapeutic study in which we infect mice with EcoHIV, then perform the adoptive transfer of viral Ag-specific iPSC-CTL. An initial experiment showed a reduction of viral replication and protection of CD4+ T cells from the damage of viral infection. The experiments are still ongoing.
Q2. The authors detect a high frequency of IL-2 production within their SL9-specifc CTL and iPSC-CTL. Is this ever observed in untransduced antigen specific normal CD8+ T cells in the EcoHIV model? In the model shown how are transferred SL9-specifc iPSC-CTL maintained? Do they require autocrine IL-2 or is there a role for CD4+ t cell derived IL-2 after infection? What is the half-life of the transferred cells?
Answer: We did not observe a high frequency of IL-2 production within untransduced Ag-specific normal CD8+ T cells in the EcoHIV model. The SL9-specific iPSC-CTL maintained in vitro by stimulation with anti-CD3 and anti-CD28 antibodies, and then with rIL-2. In vivo, after adoptive transfer, these cells require IL-2 for surviving and playing effector function; autocrine IL-2 from CD4+ or CD8+ T cells is needed. We detected a drop of cell number of the transferred iPSC-CTL (GFP+) after two weeks, assuming that the half-life of the transferred cells is around 15-20 days.
Q3. The authors focus on suppression of HIV-1 replication in peritoneal macrophages. What is the impact on infected CD4+ T cells?
Answer: The adoptive transfer of viral Ag-specific iPSC-CTL also suppresses PD-1 expression on CD4+ T cells, and protects CD4+ T cells from viral infection-induced cell death.
Q4. In figures 2, 3, 6 and 7 can the authors clarify what is meant by ‘ 3 duplicate experiments’ 3 independent experiments etc and describe precisely what id represented by the bars around the individual values (median with IQR, mean with SD etc?). The precise number of mice tested in each group or replicate experiment should be stated.
Answer: Sorry for the mistakes. We correct them.
Q5. In figure 5C the authors should either quantify the data in a graph or depict the number of GFP positive events on the figure.
Answer: We update the Fig.5 and include the quantitation (Fig. 5D).
Reviewer 2 Report
The study describes generation of CD8+ cytotoxic T lymphocytes (CTLs) from mouse iPCS followed by transduction with a cassette for the expression of TCR alfa and beta chains specifically recognizing SL9 epitope in HIV-1 Gag. The generated SL9-specific iPSC-CTLs were tested in vitro and in vivo in EcoHIV mouse model in terms of cell functionality and protection against HIV-1. The cells were first characterized in vitro and then adoptively transferred to mice challenged with EcoHIV-1. SL9-iPSC-CTLs were similarly active against HIV as CTLs transduced with SL9, however, elicited longer persistence time in vivo. In addition to proviral DNA decrease, the transfer SL9-iPSC-CTLs reduced the level of PD1 expression on CD4 lymphocytes, a marker of lymphocyte exhaustion during HIV infection. This is an interesting study and can be recommended for publication in Viruses. The manuscript is well written and structured.
The only recommendation is to check the English usage throughout the text to remove any neglects and abortive phrases. For example, lines 128-129 “We did not detect any HIV DNA copy in uninfected mice, but substantially revealed in the spleen, liver and kidney (Fig. 3A)”
Also the images in Figure 5C are very dim to see GFP+ cells.
Author Response
Q1. The only recommendation is to check the English usage throughout the text to remove any neglects and abortive phrases. For example, lines 128-129 “We did not detect any HIV DNA copy in uninfected mice, but substantially revealed in the spleen, liver and kidney (Fig. 3A).
Answer: Sorry for the mistakes. We perform the corrections.
Q2. The images in Figure 5C are very dim to see GFP+ cells.
Answer: We improve the Fig. 5, and include the quantitation of GFP+ cells (Fig. 5D).